# Line Laser Scanning Combined with Machine Learning for Fish Head Cutting Position Identification

**DOI:** 10.3390/foods12244518

**Published:** 2023-12-18

**Authors:** Xu Zhang, Ze Gong, Xinyu Liang, Weichen Sun, Junxiao Ma, Huihui Wang

**Affiliations:** 1School of Mechanical Engineering & Automation, Dalian Polytechnic University, Dalian 116034, China; zhangxu_dlut@163.com (X.Z.); 18252648000@163.com (Z.G.); sswwcc1999@126.com (W.S.); dlpujxxy@126.com (J.M.); 2School of Food Science & Technology, Dalian Polytechnic University, Dalian 116034, China; y15714110553@126.com; 3National Engineering Research Center of Seafood, Dalian 116034, China

**Keywords:** fish, head cutting position, linear laser scanning, identification model

## Abstract

Fish head cutting is one of the most important processes during fish pre-processing. At present, the identification of cutting positions mainly depends on manual experience, which cannot meet the requirements of large-scale production lines. In this paper, a fast and contactless identification method of cutting position was carried out by using a constructed line laser data acquisition system. The fish surface data were collected by a linear laser scanning sensor, and Principal Component Analysis (PCA) was used to reduce the dimensions of the dorsal and abdominal boundary data. Based on the dimension data, Least Squares Support Vector Machines (LS-SVMs), Particle Swarm Optimization-Back Propagation (PSO-BP) networks, and Long and Short Term Memory (LSTM) neural networks were applied for fish head cutting position identification model establishment. According to the results, the LSTM model was considered to be the best prediction model with a determination coefficient ($R^{2}$) value, root mean square error (RMSE), mean absolute error (MAE), and residual predictive deviation (RPD) of 0.9480, 0.2957, 0.1933, and 3.1426, respectively. This study demonstrated the reliability of combining line laser scanning techniques with machine learning using LSTM to identify the fish head cutting position accurately and quickly. It can provide a theoretical reference for the development of intelligent processing and intelligent cutting equipment for fish.

## 1. Introduction

The main process in fish processing includes scaling, gutting, cleaning, and head/tail cutting, where head removal is an important part of cutting planning and directly affects the processing quality and meat yield [1]. The main types of head cutting processes are manual and mechanical [2]. The manual method is time-consuming and laborious, with low processing efficiency and high skill requirements for the processors [3], which cannot be adapted to the needs of short-term and high-volume production of bulk fish. Due to the biodiversity of fish, even the size of the heads of the same specification and the same batch of raw material varies greatly, and existing mechanical cuts are processed according to a pre-set cutting position [2], which is unsatisfactory in terms of reducing meat yield and non-compliance with handling. To be specific, if the processing volume is set too large, it will lead to a lower meat yield and waste; if the processing volume is set too small, the cutting tool will be easily damaged by cutting at the gill cover, resulting in cutting failure and even causing failure of the entire equipment operation. Therefore, how to achieve automatic identification of the fish head position so as to control the accurate cutting of the fish head is a problem that needs to be urgently solved for flexible, intelligent, and efficient processing of bulk fish.

Dynamic 3D reconstruction is one of the important technologies for fast and accurate contour feature identification in advanced manufacturing, and food and agricultural product processing [4,5]. The acquisition of the measured object contour data is mainly realized by means of computer vision, lasers, ultrasound, NMR, and X-rays. Among them, single/binocular vision usually establishes the inference rules by capturing image information such as shape, texture, and color, and has been widely used in food size and volume measurements [6]. Ultrasound, NMR, and X-rays are mostly used to probe the interior of objects, such as the application of NMR technology in fresh meat quality evaluation, X-ray for dark field imaging in medicine, etc. Laser sensors are suitable for high-volume installation and have been effectively applied in shape monitoring in animal husbandry and in planting and pruning of agricultural crops, which are not susceptible to the influence of the external lighting environment [7,8,9,10]. Due to the technical characteristics of industrial fish head removal processing, the high measurement accuracy required, the complex and variable processing environment, and the large installation capacity of the equipment, this study selected laser scanning technology for fish surface data information collection. In addition, to achieve accurate cutting control of the fish head, it is necessary to build a reliable mathematical model for predictions. The traditional single-factor linear regression prediction method is simple in principle and mature in technology, but it lacks a self-learning capability and is difficult to achieve accurate descriptions of complex non-linear models. In recent years, machine learning has performed superiorly in the field of intelligent control, gradually replacing traditional prediction methods [11,12]. Least Squares Support Vector Machines (LS-SVMs) overcome the shortcomings of artificial neural networks and SVM to achieve global optimality with the principle of structural risk minimization, which is more widely used in the field of food testing, and has been used in the identification of salmon moisture content, the prediction of acetic acid content in beer, and the prediction of changes in freshness indicators during the refrigeration of trout fillets with high accuracy [13,14,15,16]. Particle Swarm Optimization-Back Propagation (PSO-BP) neural networks addresses the problem where traditional Back-Propagation (BP) neural networks easily to fall into local optima and has great advantages in weight and bias initialization, learning rate adjustment, and convergence speed, and is widely used in agriculture and has achieved good results in tea water content prediction, coliform amount prediction, grain yield prediction, and so on [17,18,19]. A Long and Short Term Memory (LSTM) neural network is a supervised neural network that addresses the long-term dependency problem in Recurrent Neural Networks (RNNs); as a non-linear model, LSTM can be used as a complex non-linear unit to construct larger deep neural networks, mostly for the quantitative analysis of important elements of samples, prediction of important components, and so on [20,21]. The above studies provide a lot of references and support for the present study on the identification of fish head cutting position using fish body surface feature information and the establishment of reliable models.

The aim of this study was to explore the feasibility of achieving automatic fish head position identification by using a constructed fish surface contour laser scanning system and to propose a rapid fish head identification method based on 3D contour information so as to provide fast and accurate cutting path planning for automated and intelligent head removal processing. The main research contents of this paper are as follows: (1) constructing a laser scanning system for fish body data information to realize the automatic acquisition of contour information on the outer surface of the fish body; (2) proposing a data validity discrimination and filtering method suitable for feature extraction of contour information of the fish body radial section; (3) taking the dimensionality reduced feature values as inputs and the fish head cutting position as outputs, MPR-, LS-SVM-, and LSTM-based fish head ideal cutting position identification models were established to achieve accurate identification of the fish head cutting position.

## 2. Materials and Methods

### 2.1. Materials

This study was carried out on crucian carp. The experimental samples were purchased from Dalian Wholesale Fish Market, China. A total of 204 crucian carp were randomly selected and placed in a thermostat with ice for rapid transportation back to the laboratory. The parameters of the fish body shape were defined as shown in Figure 1; the distance between the mouth and the trailing edge of the gill cover was defined as the head length (Figure 1a), the maximum distance from the front of the mouth to the end of the tail fin was defined as the total length (Figure 1b), and the maximum distance between the dorsal and ventral parts of the fish was defined as the maximum width (Figure 1c), setting the cut line passing through this location as the ideal cut line for the head; the height of the highest point of the fish to the level of the conveyor belt when the fish was placed horizontally was defined as the maximum thickness (Figure 1d). The statistics of the manual measurements of the 204 samples are shown in Table 1.

### 2.2. Data Acquisition and Pre-Processing

#### 2.2.1. Data Acquisition System

A fish data information laser scanning system was used to collect 3D contour information of the fish body. As shown in Figure 2b, the system mainly consisted of a line laser scanning sensor (LLT-2600 scanContral2D/3D, Micro-Epsilon, Ortenburg, German), a drive mechanism, a dark box, and a data processing unit. The laser scanning sensor scanned 640 points at a time, with a scanning frequency of 300 Hz, and outputted the measurement results via Ethernet (Modbus TCP protocol). The drive mechanism consisted of a conveyor belt and a drive unit, which can transport the material horizontally and linearly under the drive of a servo motor, and the motor speed was set at 6.6 rpm. The dark box was equipped with two groups of strip light sources, which were placed on both sides of the dark box, as shown in Figure 2b; the data processing unit was controlled by a computer, which was used to realize pre-processing such as data segmentation and filtering of the collected raw fish body information (Figure 2c).

#### 2.2.2. Data Acquisition Process

The laser sensor was calibrated to send a laser ray vertically downwards, with the line laser perpendicular to the conveyor belt transport direction. The laser scanning schematic is shown in Figure 3a,b; the intersection points between the vertical line of the laser source and the horizontal plane where the conveyor belt surface was located was defined as the O point, the height was in the Z direction, the laser direction was defined as the Y direction, and the conveyor belt conveying the reverse direction was defined as the X direction. During data acquisition, the head of the fish was orientated in the same direction as the movement direction, and when the sample triggered the timing procedure, the laser sensor started scanning the fish to obtain the 3D point cloud information of the surface contour of the fish. Due to the huge amount of real-time data obtained by laser scanning, the sampling frequency was set to 0.42 Hz without affecting the accuracy of the calculation, and the obtained contour information was actually a number of point cloud data containing the contour information of the fish body cross-section (Figure 3c), which was stored in the form of an array.

#### 2.2.3. Data Validity Discernment

As shown in Figure 3b, since the width of the line laser is larger than the fish it scans, the interference data formed on the surface of the conveyor belt on both sides of the fish body will inevitably be collected simultaneously during the process of acquiring the fish body contour data. In this study, the threshold segmentation method was used to remove the interfering data while retaining useful information on the surface of the fish body for subsequent processing. The calculation process of the threshold segmentation method is as follows.
(1)$$\left[M\right]=\left\{\begin{array}{c} \begin{array}{cc} \left[M_{1},M_{3}\right], & \Delta h_{1}<T,\Delta h_{2}<T \end{array}\\ \begin{array}{ccc} \left[M_{2}\right], & & \Delta h_{1}>T,\Delta h_{2}>T \end{array} \end{array}\right.$$
(2)$$∆h_{1}=\left|M_{1}-M_{2}\right|,$$
(3)$$∆h_{2}=\left|M_{2}-M_{3}\right|$$
where $\left[M\right]$ represents the array after threshold segmentation. $\left[M_{2}\right]$ represents the array of radial sections of the fish body. $\left[M_{1}\right]$ and $\left[M_{3}\right]$ represent the radial cross-sectional array of the conveyor belt on both sides of the fish body. T represents the segmentation threshold. $\Delta h_{1}$ represents the absolute value of the difference between adjacent elements’ height value of the right endpoint of $\left[M_{1}\right]$ and height value of the left endpoint of $\left[M_{2}\right]$. $\Delta h_{2}$ represents the absolute value of the difference between adjacent elements’ height value of the right endpoint of $\left[M_{2}\right]$ and height value of the left endpoint of $\left[M_{3}\right]$.

#### 2.2.4. Data Filtering

The absorption and reflection of light by the fish itself, the influence of external lighting, and the vibration of the conveyance mechanism during movement can cause high-frequency fluctuations in the data and form noise [22]. Since the Kalman filter has a good suppression effect on the random fluctuation noise generated by the data, and the median filter can reduce the fluctuation range of the data, the Kalman filter and the median filter were adopted to denoise the data. In this study, the covariance of the system noise of the Kalman filter was set to 0.0001, the covariance of the measurement noise was set to 0.1, the covariance of the system noise was set to 1, and the left and right rank of the median filter were set to 2 and −1, respectively.

### 2.3. Fish Head Cut Position Identification

#### 2.3.1. Feature Extraction

As shown in Figure 3d, the fish head cut position is on the contour line composed of the highest points of the radial section data of the fish body, namely the boundary between the abdomen and the back of the fish body, which is defined as the ventral–dorsal demarcation line of the fish body in this study. The sampled data on the ventral–dorsal demarcation line of the fish body were taken as input and the real value of the fish head cut position was taken as output to construct the fish head cut position identification model and to achieve the prediction of the fish head cut position. The volume of data on the ventral–dorsal demarcation line is large, and there is a strong correlation between some data points, making it a large amount of redundant information and affecting the calculation accuracy. Principal Component Analysis (PCA) is an unsupervised machine learning algorithm that can transform multiple variables into a few composite variables, eliminating redundant information and reducing computational effort [23]. In this study, PCA was chosen to reduce the dimensionality of the ventral–dorsal divide, using a few principal components instead of the entire ventral–dorsal divide data. As shown in Equations (4)–(6), the collected ventral–dorsal dividers were transformed by the Z-score method to standardize them, as a way to eliminate the difference in magnitude and the difference in order of magnitude between different indicators. The correlation coefficient matrix between the independent variables was solved by using a standardized data matrix, and the characteristic roots were obtained according to the characteristic equation of the correlation coefficient matrix, as shown in Equation (8), with the cumulative contribution of the variance at 95% to determine the extracted principal components for subsequent studies. As shown in Equations (9) and (10), the indicator coefficient matrix of each principal component was obtained according to the component matrix and multiplied with the standardized data matrix to obtain the principal component values.
(4)$${\bar{z}}_{j}=\frac{\sum_{i=1}^{n}z_{ij}}{s_{j}},s_{j}=\sqrt{\frac{\sum_{i=1}^{n}{\left(z_{ij}-{\bar{z}}_{j}\right)}^{2}}{n-1}}$$
(5)$${\tilde{z}}_{ij}=\frac{z_{ij}-{\bar{z}}_{j}}{s_{j}}$$
(6)$$Z_{ij}=\frac{z_{ij}-{\bar{z}}_{j}}{s_{j}},D=\frac{{Z_{ij}}^{T}Z_{ij}}{n-1}$$
(7)$$\left|D-\lambda I_{p}\right|=0$$
(8)$$\frac{\sum_{j=1}^{m}\lambda_{i}}{\sum_{j=1}^{p}\lambda_{i}}\geq 0.95$$
(9)$$U_{i}=P_{i}/\sqrt{\lambda_{i}}$$
(10)$$F_{i}=\sum_{i=1}^{n}U_{i}\times ZX_{i}$$
where ${\bar{z}}_{j}$ represents the mean value of the ventral–dorsal divide, $s_{j}$ represents the standard deviation of the ventral–dorsal divide, ${\tilde{z}}_{ij}$ represents the standardized data, Z_ij_ represents the standardized matrix combined by column, D represents the correlation coefficient matrix, λ represents the eigenvalues of this matrix, P_I_ represents the component matrix, U_I_ represents the index coefficient matrix, and F_i_ represents the principal component values of the ventral–dorsal divide of the fish.

#### 2.3.2. Establishment of Fish Head Cut Position Identification Models

The principal component values of 204 fish body samples after PCA dimensionality reduction were used as model inputs and the length of the fish head was the output to construct the LS-SVM, PSO-BP, and LSTM models for the identification of the cutting position of the fish head. Among them, a total of 154 samples were randomly selected as the training set, and the remaining 50 samples were used for the testing set.

##### LS-SVM Model

The LS-SVM maps linearly indistinguishable data in space to a high-dimensional feature space by means of a constructed kernel function that makes the data divisible in the feature space [24,25]. Given that the Gaussian radial basis function (RBF) is capable of non-linear mapping and has fewer parameters, the RBF kernel function was chosen to construct the LS-SVM model in this study. The penalty parameter sig2 and the radial basis kernel parameter gam are two important parameters of the RBF kernel function, which are closely related to the accuracy and generalization ability of the model [26]. In this study, the particle swarm algorithm (PSO) was used to find the best values for the above two parameters, and the maximum number of iterations was taken as 100, with the search range of sig2 being 0.1 to 100 and the search range of gam being 0.01 to 100.

##### PSO-BP Model

BP neural network is an error backpropagation algorithm, and its learning process can be summarized as signal forward propagation, error backpropagation, weights, and threshold update. The network includes input, hidden, and output layers. Using the gradient descent method, the weights and thresholds between different network layers are adjusted inversely by comparing the model output values with the expected values to reduce the error along the gradient direction [19]. The approximate solution that satisfies the error accuracy is sought through several iterations, and its structure is shown in Figure 4.

In a traditional BP network, it is easy to fall into local optimal solutions in the training process, which makes the final model accuracy too low [17]. To avoid this problem, this study used the PSO algorithm with global search ability to optimize the network weight of the BP neural network. Figure 5 shows the structure of the PSO-BP neural network built in this study, where the inputs are three independent variables, the number of hidden layers was ten layers, and the parameters of the neural network were set as follows: the number of iterations was 1000, the training objective was 0.00001, and the learning rate was set to 0.09. The PSO parameters were set as follows: learning factor $c_{1}=c_{2}=2$, inertia weights$\;\omega_{max}=0.9$ and $\omega_{min}=0.3$ with a maximum iteration of 200.

##### LSTM Model

The LSTM model is a special type of Recurrent Neural Network (RNN), consisting of memory blocks that add input and output channels to the hyperbolic tangent function (tanh), which can correlate the feature data of each fish with each other and analyze their non-linear relationships [27]. The cell structure of the LSTM model is shown in Figure 6, where$\;f_{t}$, $i_{t}$, and $o_{t}$ denote the forgetting layer, input layer, and output layer, respectively; $x_{t}$ is the input of the current cell; $C_{t-1}$ and $h_{t-1}$ are the outputs of the last network cell; W and V are the weight matrices; b is the bias term; and σ is the sigmoid function layer. The LSTM network used in this study has an input layer with three inputs, a hidden layer activated by the tanh function, one forgetting layer, one fully connected layer, and one output layer. The gradient threshold size used was 1, the initial learning rate was 0.005, and the maximum number of iterations was 200.

### 2.4. Model Evaluation Metrics

In order to verify the identification effect of the model, the coefficient of determination ($R^{2}$), the root mean square error (RMSE), the mean absolute error (MAE), and the relative analysis error (RPD) were used as evaluation indicators of the prediction model. $R^{2}$ is an important parameter in model evaluation, and usually $R^{2}$ > 0.82 indicates that the method can be used for practical applications. $R^{2}$ > 0.9 indicates that the model has excellent fish head cutting position identification. Smaller RMSE values represent better model prediction performance [28]. MAE can better reflect the actual situation of prediction value error, and a smaller value represents a smaller prediction error [28,29]. RPD can intuitively reflect the prediction ability of the model; when RPD > 1.4, the model reliability is poor. When the RPD > 2.5, the model prediction is accurate and reliable [30,31]. The formulas of the above evaluation indexes are as follows:(11)R2=1−∑i=1nyi−yi^2∑i=1nyi−y−i2
(12)$$RMSE=\sqrt{\frac{\sum_{i=0}^{n}{\left(y_{i}-{\overset{̑}{y}}_{i}\right)}^{2}}{n}}$$
(13)$$MAE=\frac{1}{n}\sum_{i=1}^{n}\left|y_{i}-{\overset{̑}{y}}_{i}\right|$$
(14)$$RPD=\frac{SD}{RMSEP}=\sqrt{\frac{1}{1-{R_{p}}^{2}}}$$

## 3. Results

### 3.1. Data Pre-Processing

#### 3.1.1. Data Segmentation

As shown in Figure 7a, the initial point cloud data obtained after laser scanning included the conveyor belt contour and the fish surface contour point cloud data. The maximum horizontal height of the conveyor belt contour was 250.32 mm, and this part was removed according to the method described in Section 2.3.2 to achieve partial dimensionality reduction of the data and eliminate the interference information irrelevant to the fish contour (Figure 7b).

#### 3.1.2. Data Filtering

The results of the original data filtering of the radial cross-section profile of the fish body are shown in Figure 8. As can be seen in Figure 8a, the original data had more obvious noise and data fluctuation, and the radial cross-section profile of the fish body was not accurate. After adding the Kalman filter, the large-scale noise generated by system vibration was basically removed, indicating that it has a good suppression effect on the noise generated by random fluctuations in this study (Figure 8b), and the radial profile curve was further refined. But, due to the existence of data fluctuations, the profile curve was still not smooth and complete. After adding the median filter, the range of fluctuations of the original data was reduced (Figure 8c), and the radial profile curve gradually became continuous and complete. When the data were subjected to Kalman and median filters in turn, the contour curve became smooth and the high-frequency noise was obviously improved (Figure 8d), which is closer to the real contour curve. Therefore, in this study, the Kalman filter and the median filter for the pre-processing of the radial contour of the fish body were used together.

### 3.2. Fish Head Cut Position Identification

#### 3.2.1. Extraction of Ventral–Dorsal Demarcation Line

For the fish body, the thickness of the ventral–dorsal part of the fish decreased slowly, and the thickness of the tail part of the fish decreased rapidly. The maximum point of the thickness of the fish was at the ventral–dorsal part, and the position of the head cut was before the maximum point of the thickness. The ventral–dorsal demarcation line of a fish is a cloud of points consisting of the maximum values of all radial cross-section heights of the fish body. The ventral–dorsal demarcation lines of 204 fish bodies are shown in Figure 9. It illustrates that although the size of the samples was not uniform, the corresponding ventral–dorsal demarcation lines had the same trend of changes, which increased rapidly and reached the maximum value before about one-third of the total length, and then began to decrease. As shown in the figure (Figure 9), the ventral–dorsal demarcation line could reflect the change law of the radial thickness of the fish body, which can be used as the basis for fish head cutting. Taking the ventral–dorsal demarcation line as input, three supervised machine learning methods (LS-SVM, PSO-BP, and LSTM) were used to train and predict the model, respectively, to achieve the identification of the fish head cutting position.

#### 3.2.2. Data Dimensionality Reduction of Abdominal and Dorsal Dividing Lines

In order to reduce the volume of data for the identification of fish head cut features and to achieve rapid identification of fish head cut locations, PCA was used to reduce the dimensionality of the abdominal and dorsal dividing lines. Factor analysis was performed on the measured ventral–dorsal demarcation lines of the 204 samples, and the results of the obtained feature values are shown in Table 2. The cumulative variance contribution of the first three principal components reached 95.080%, indicating that the first three principal components can significantly reflect 95.080% of the information of the original data. In addition, it can also be seen from the gravel plot shown in Figure 10 that the trend of the first, second, and third eigenvalues was more obvious, and from the fourth eigenvalue onwards, the trend of the eigenvalues tended to be stable, so the first three principal components were taken for subsequent modeling.

#### 3.2.3. Fish Head Cutting Position Identification Model

Taking the three principal component values obtained as the model input and the actual cut position of the fish head as the model output, all 204 samples were randomly divided into two groups, one group of 154 samples as the training set and the other group of 50 samples as the test set, to construct the LS-SVM, PSO-BP, and LSTM fish head cutting position recognition models.

##### LS-SVM Model

In order to obtain an LS-SVM model with high prediction accuracy and stability, the two parameters of the RBF kernel function in the model, penalty coefficient sig2 and kernel parameter gam, need to be optimized. The results of parameter search using PSO showed that the LS-SVM model had the best recognition effect when the penalty parameter sig2 = 0.01 and the optimal kernel parameter gam = 15.3284; the fitness curve is shown in Figure 11, from which, after 29 iterations, the fitness curve started to smooth out and the validation parameters reached the optimal solution. The recognition results of the LS-SVM model for the fish head cutting position are shown in Figure 12. The $R_{c}^{2}$, RMSE, and MAE of the training set were 0.9125, 0.2622, and 0.1857, respectively, and the $R_{p}^{2}$, RMSE, MAE, and RPD of the test set were 0.9094, 0.8548, 0.6123 and 2.4041, respectively. The results showed that its $R_{p}^{2}$ was more than 0.9, which indicates that the LS-SVM model has good generalization ability and performance, and RMSE and MAE were low, which represents fewer outliers and errors in the results predicted by the model, but its RPD values were less than 2.5, indicating that the model is not stable enough and has limited identification capability.

##### PSO-BP Model

The connection weights of each layer of the BP neural network were encoded into particles, and the PSO algorithm was used to search for the optimal network weights within the set number of iterations, in which the population size of the particle swarm was set to be 20 to prevent overconsumption of computational resources. In order to avoid the particle speed growth being too fast or too slow, the value of the speed was set to [−1,1], ensuring that the solution is within a reasonable range of the particle position to avoid overstepping the boundaries, and the position was set to [−2,2]. The training results of the PSO-BP neural network for fish head cutting position identification are shown in Figure 13. After the computation of the PSO-BP model, the model’s recognition results of the fish head cutting position are shown in Figure 14. The $R_{c}^{2}$, RMSE, and MAE of the training set were 0.9168, 0.5203, and 0.3277, respectively, and the $R_{p}^{2}$, RMSE, MAE, and RPD of the test set were 0.9295, 0.5126, 0.3143, and 2.513, respectively. Compared with the LS-SVM model, its larger $R_{p}^{2}$ value represents a better model performance, its smaller RMSE and MAE, represent a further reduction in the error of the PSO-bp model prediction, and its RPD was more than 2.5, which indicates that its recognition has strong reliability and is suitable for the recognition of the ideal cutting position of fish heads.

##### LSTM Model

During the LSTM training, the forgetting, input, and output layers were activated by sigmoid functions, and the entire output data range was transformed in the [0,1] interval to keep the data normalized. In the model-building process, if a neuron parameter produces large volatility, the overall fit of the model will be biased towards that neuron; therefore, during the training process, to reduce the impact of overfitting on the prediction model, dropout regularization was added, and the dropout rate was taken as 0.2. As can be seen from Figure 15, during 200 iterations, there was a brief upward trend in the loss function of the training and test sets, followed by a rapid decline, which slowed down during the subsequent iterations, indicating that the LSTM output values fit the true values better and better. The results of the LSTM model for the identification of fish head cutting positions are shown in Figure 16. The $R_{c}^{2}$, RMSE, and MAE of the training set were calculated by the LSTM model to be 0.9705, 0.1964, and 0.1477, respectively, and the $R_{p}^{2}$, RMSE, MAE, and RPD of the test set were 0.9480, 0.2957, 0.1933, and 3.1426, respectively. Among the analyzed models, the LSTM model’s $R_{p}^{2}$ reached up to 0.9480, with the best generalization ability and performance, and it had a lower RMSE and MAE, which represents minimum and stable prediction error, and a larger RPD value of more than 2.5, which indicates that its identification has strong reliability and is suitable for the recognition of the ideal cutting position of fish heads.

In the present study, the interference information and random fluctuation noise in the original data were processed by threshold segmentation and Kalman and median filtering, respectively, which can accurately restored the radial profile curve of the sample. The principal component value of the ventral–dorsal demarcation lines after PCA dimensionality reduction can be used as the feature information for identifying the cutting position of the fish head. In the established MPR, LS-SVM, and LSTM fish head cutting position identification models, the overall performance of the LSTM model was the best: the error between the predicted value and the actual value of the model was the smallest, and the reliability was also high. This shows that the line laser scanning technology combining with machine learning has the potential for fish head cutting position recognition. At the same time, the identification based on the information of the ventral–dorsal dividing line proved to be effective. The ventral–dorsal demarcation line is a line composed of the highest points in each radial profile section, which is identified by the morphological characteristics of the fish. It will not be limited to the posture and individual size, so the accuracy of the identification method based on the ventral–dorsal demarcation line will not be affected by the placement and size of the fish. The spindle-shaped and flat-hammer-shaped fishes with a similar morphology to that of the samples used in this study (crucian carp) all have a ventral–dorsal dividing line, and therefore, the method proposed in this study could have strong applicability to bulk fish with similar shapes and head-removing needs, such as grass carp, silver carp, etc., and even high value-added fish such as salmon and tuna. With a further increase in sample size, the accuracy and generalization of the identification model could also be continuously improved. In addition, due to the high precision and efficiency of line laser scanning, the application of this method is more conducive to the on-line detection of bulk fish. However, in actual large-scale fish processing, when two or more fish are stacked up and placed, it is hard to extract the ventral–dorsal demarcation line of an individual sample fish, so the method should be applied to the situation where raw materials are scanned one by one.

## 4. Conclusions

In this study, we used line laser scanning technology to nondestructively collect the surface information of the fish body, restore the outer contour information of the fish body through data processing, and extract the feature variables of the ventral–dorsal demarcation line through PCA to reduce the computation time and improve the identification ability. Using the obtained feature variables as the inputs, fish head identification models based on LS-SVM, PSO-BP, and LSTM were established. The results showed that the data processed by threshold segmentation, Kalman filtering, and median filtering can restore the radial contour of the fish body more accurately. In the model comparison experiments, the LSTM model outperformed the LS-SVM and PSO-BP model with the highest R^2^ and RPD of 0.9480 and 3.1426, and lowest RMSE and MAE of 0.2957 and 0.1933, respectively, in the test set. These results validate the potential of combining line laser scanning techniques with machine learning for fish head cutting position identification. In future studies, experiments will be conducted on more types and sizes of fish to improve the reliability and practicality of the model.

## Figures and Tables

**Figure 1 foods-12-04518-f001:**
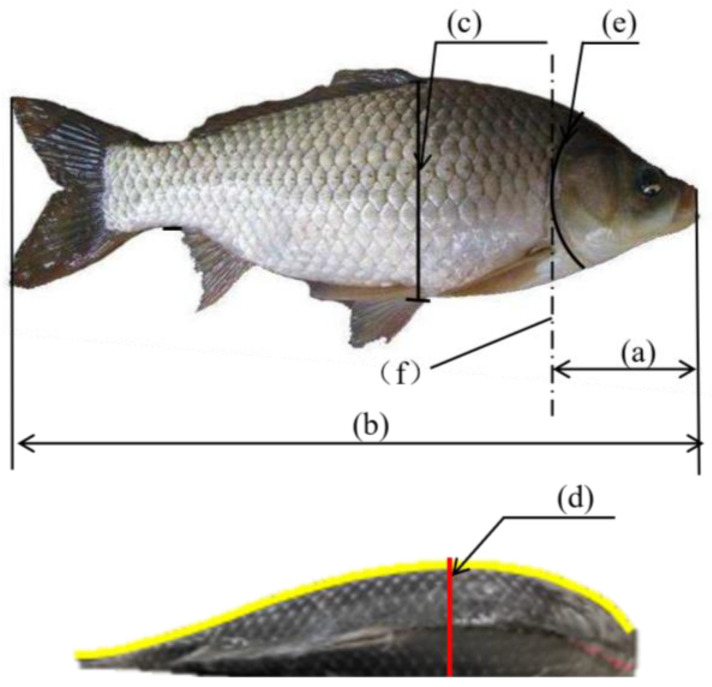
Relationship between head cutting position and fish body parameters: (a) fish head length; (b) total length; (c) maximum width; (d) maximum thickness; (e) ideal cutting line; (f) gill.

**Figure 2 foods-12-04518-f002:**
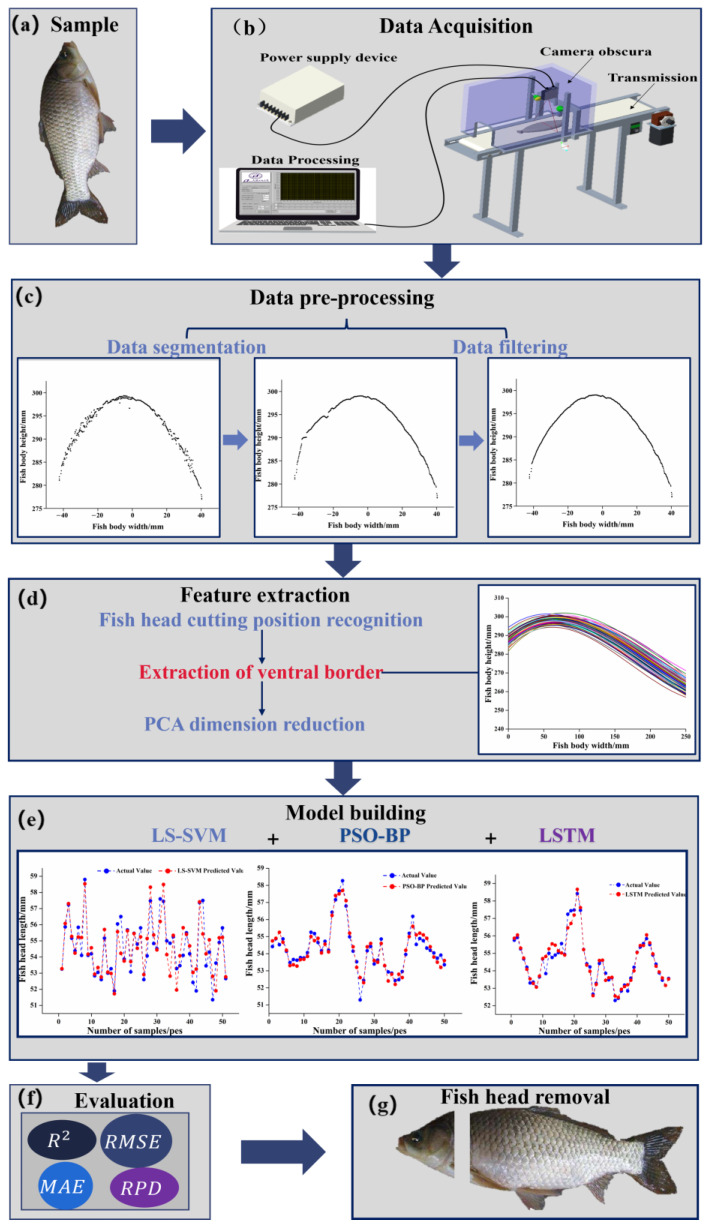
Fish head cutting position determination flow chart: (**a**) sample; (**b**) data acquisition; (**c**) data pre-processing; (**d**) feature extraction; the different colored lines represent the ventral–dorsal dividing line extracted from different samples; (**e**) model building; (**f**) evaluation; (**g**) fish head removal.

**Figure 3 foods-12-04518-f003:**
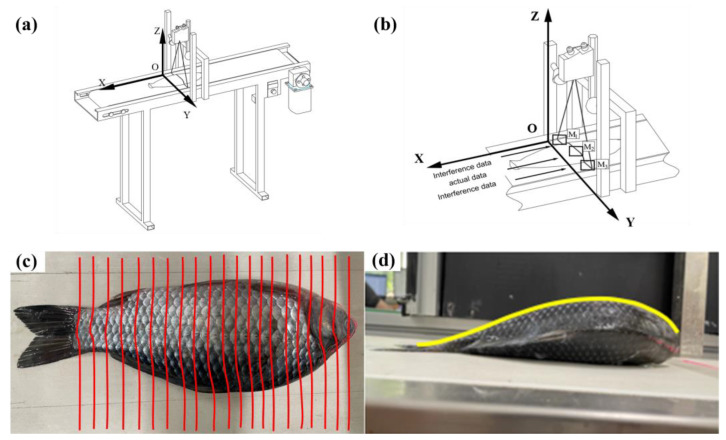
Data acquisition process: (**a**) coordinate diagram; (**b**) scanning process diagram; (**c**) radial section contour diagram; (**d**) Y projection contour diagram.

**Figure 4 foods-12-04518-f004:**
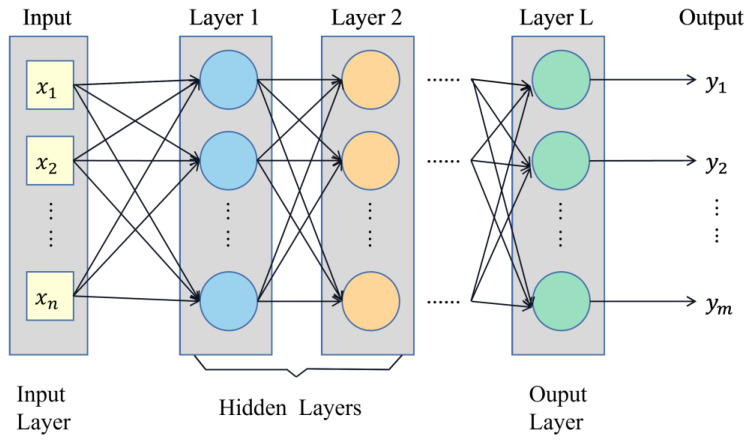
BP neural network unit structure.

**Figure 5 foods-12-04518-f005:**
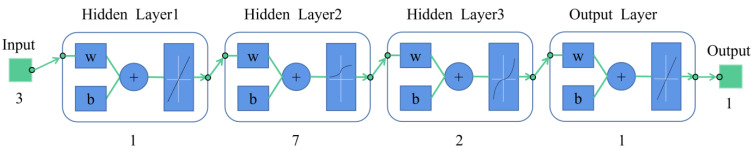
PSO-BP unit structure.

**Figure 6 foods-12-04518-f006:**
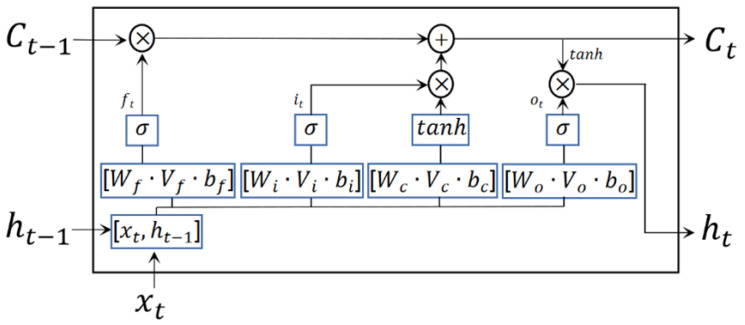
LSTM unit structure.

**Figure 7 foods-12-04518-f007:**
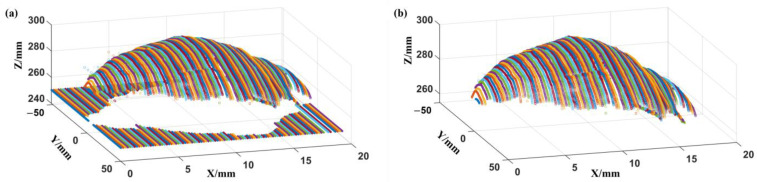
Upper surface data collection of fish: (**a**) laser point cloud data collection; (**b**) fish surface data after segmentation. The different colored lines represent the results of a single line laser scan and consist of multiple points that form a radial profile.

**Figure 8 foods-12-04518-f008:**
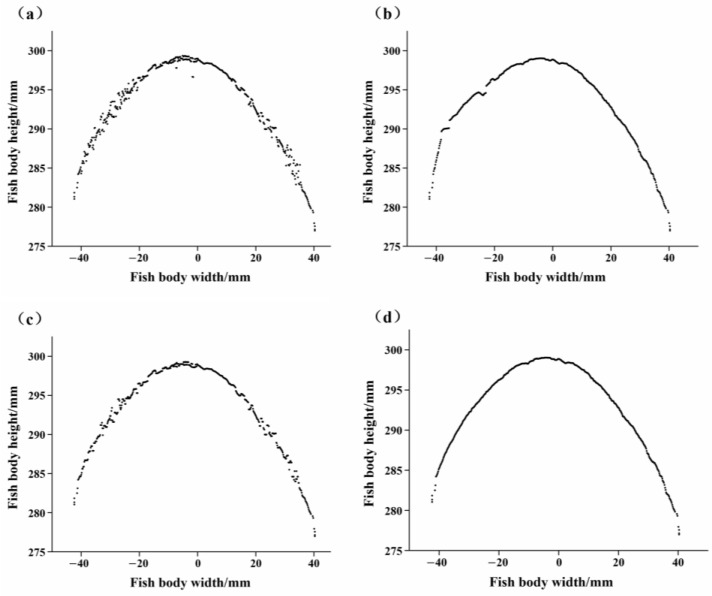
Data filtering: (**a**) original data; (**b**) Kalman-filtered data; (**c**) median-filtered data; (**d**) Kalman- and median-filtered data.

**Figure 9 foods-12-04518-f009:**
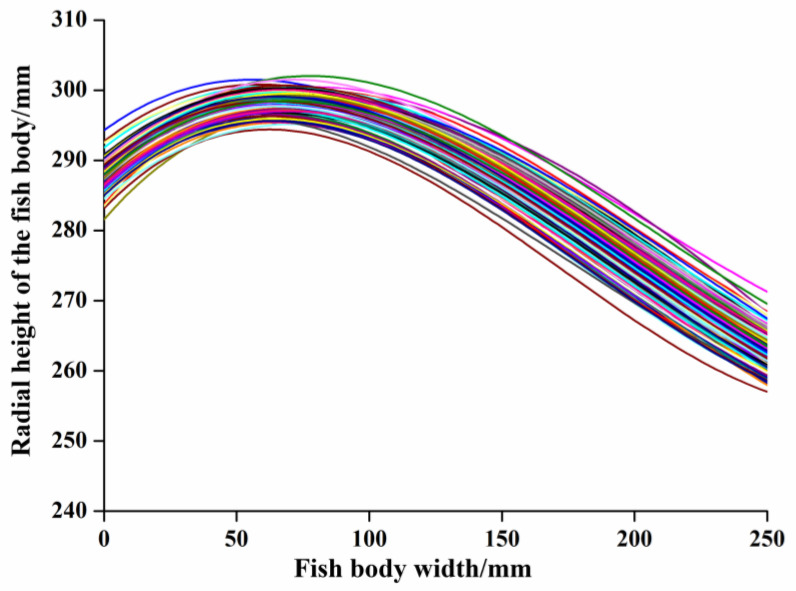
Abdominal and dorsal dividing lines. The different colored lines represent the ventral–dorsal dividing line extracted from different samples.

**Figure 10 foods-12-04518-f010:**
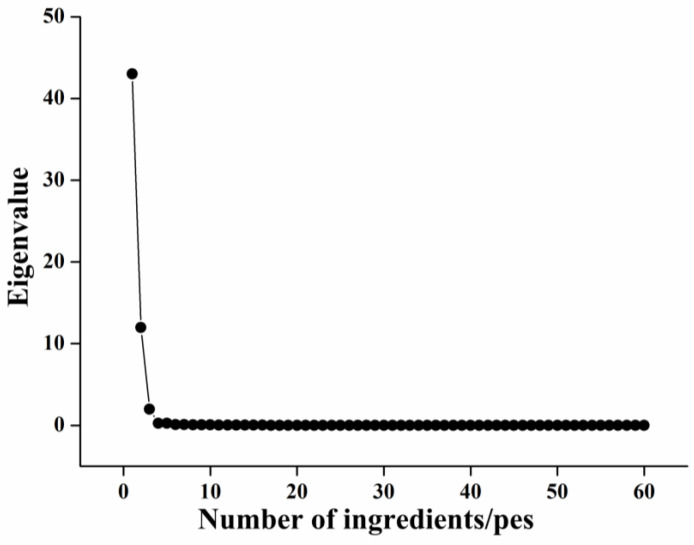
Gravel map extracted based on covariance matrix.

**Figure 11 foods-12-04518-f011:**
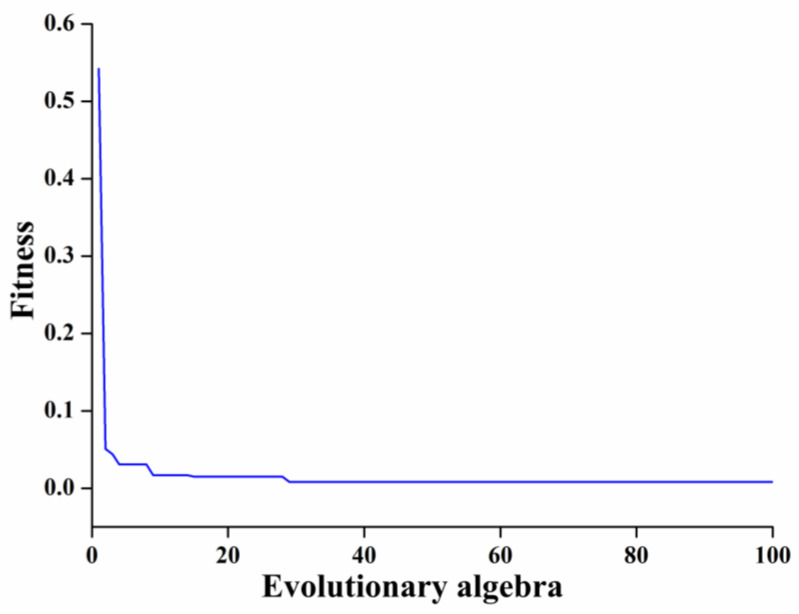
Fitness diagram.

**Figure 12 foods-12-04518-f012:**
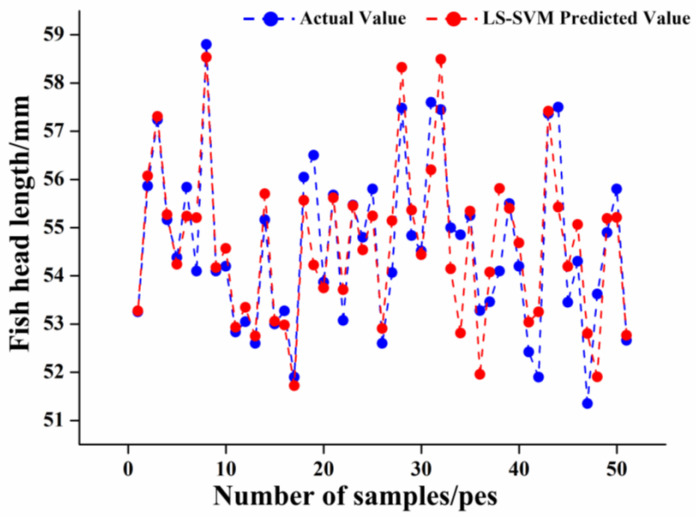
Comparison diagram of predicted and measured fish head length based on LS-SVM.

**Figure 13 foods-12-04518-f013:**
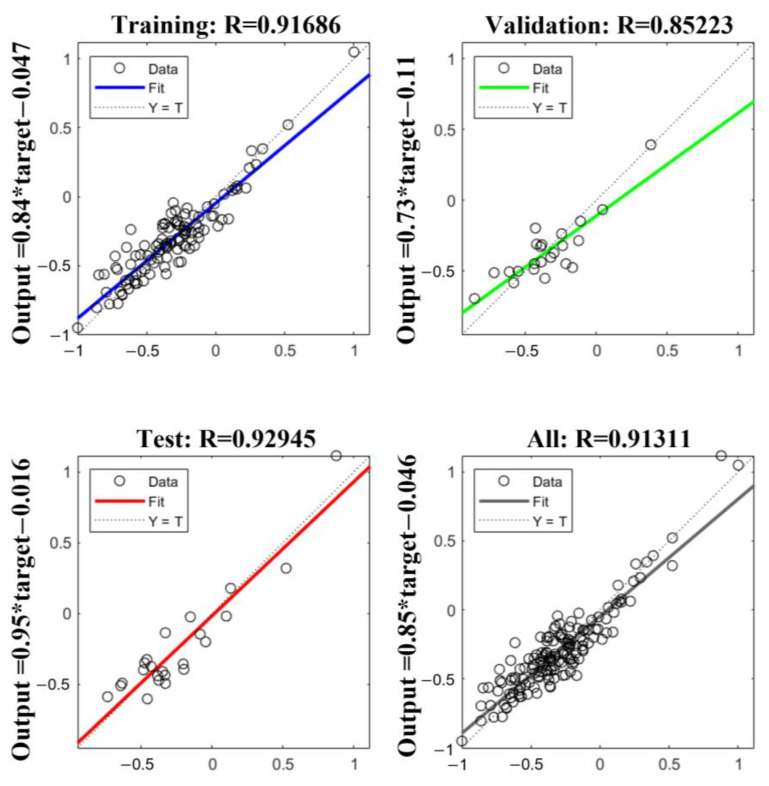
PSO-BP unit structure training results.

**Figure 14 foods-12-04518-f014:**
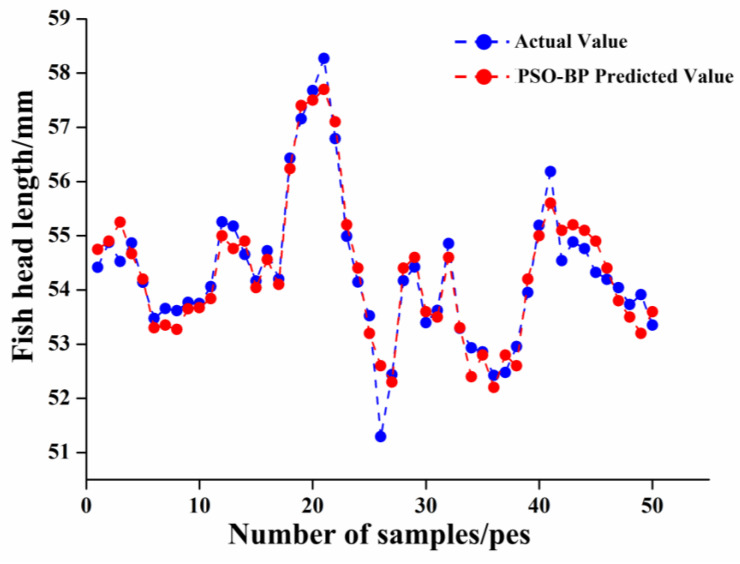
Comparison diagram of predicted and measured fish head length based on PSO-BP.

**Figure 15 foods-12-04518-f015:**
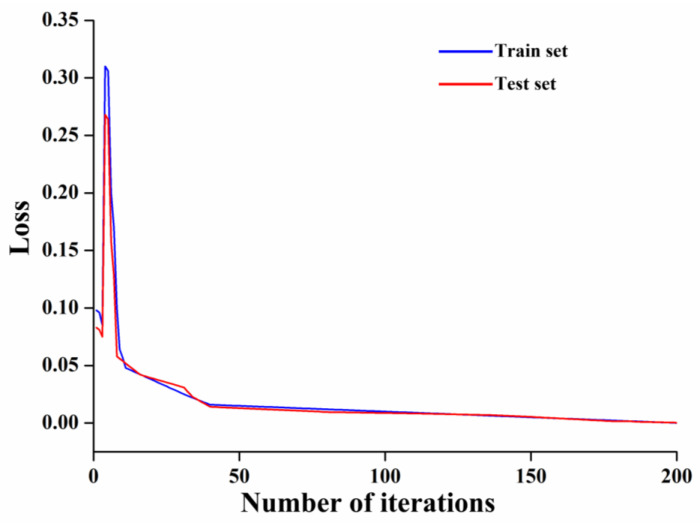
The loss function of training iterations and the target value of the LSTM model.

**Figure 16 foods-12-04518-f016:**
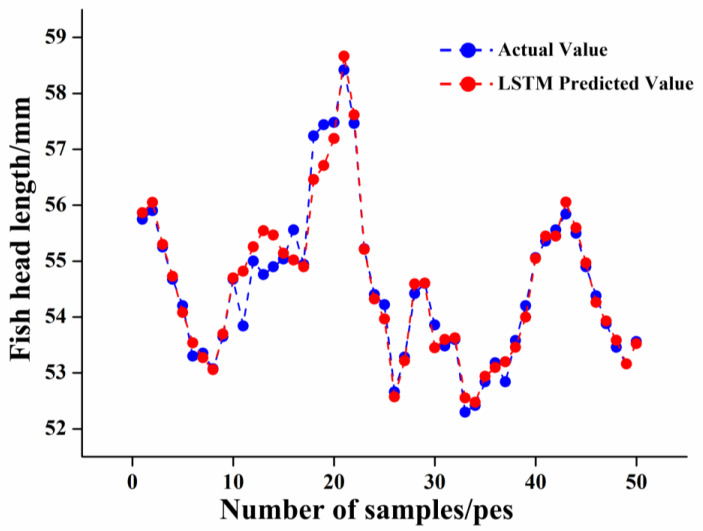
Comparison diagram of predicted and measured fish head length based on LSTM.

**Table 1 foods-12-04518-t001:** Index statistics of fish sample data.

Sample Size	Statistical Indicator	a/mm	b/mm	c/mm	d/mm	Weight/g
204	Maximum value	63.1	239.4	103.7	51.54	569
	Minimum value	50.1	200.8	94.6	45.3	473
	Mean	223.6	54.6	100.9	47.49	532.1
	Standard deviation	2.5	8.9	2.2	1.85	21.6

**Table 2 foods-12-04518-t002:** Total variance of interpretation.

Component	Initial Eigenvalue			Extraction of Squares and Loading			Rotate the Square and Load		
	Total	Variance of %	Cumulative %	Total	Variance of %	Cumulative %	Total	Variance of %	Cumulative %
1	43.031	71.718	71.718	43.031	71.718	71.718	34.740	57.901	57.901
2	11.990	19.983	91.702	11.990	19.983	91.702	16.894	28.157	86.058
3	1.999	3.3319	95.033	1.999	3.331	95.033	5.385	8.974	95.033
4	0.842	1.403	96.436						
5	0.386	0.643	97.079						
6	0.340	0.567	97.646						
7	0.261	0.434	98.080						
8	0.139	0.232	98.312						
9	0.112	0.187	98.496						
10	0.097	0.161	98.661						
		......							
204	−1.04 × 10^−15^	−1.74 × 10^−15^	100.00						

## Data Availability

Data are contained within the article.

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
