# Peer review of "Line Laser Scanning Combined with Machine Learning for Fish Head Cutting Position Identification"

_foods, 2023, doi:10.3390/foods12244518_

Round 1

Reviewer 1 Report

Comments and Suggestions for Authors

I reviewed the manuscript titled “Research on Fish Head Cutting Position Recognition Model Based on Line Laser Scanning”. The manuscript is well written and contributes to the field

Title: title should be revised, particularly, research on

Abstract

Conclusions should in included in this section

Introduction

Provide the space between citation and end of the sentence. This must be revised throughout the manuscript

Lines 84 and 96: objectives are repeated. Authors should revise this section

Figure 2e: quality must be improved

2.3.2.2 PSO-BP model: provide citation

Lines 194, 196: it must be equations 6 and 7

Results are appropriate with models and their accuracy with statistical models. However, how practical it is for a large-scale industry.

Are the fishes need to arrange in a row to cut their head? Zigzag (fishes are coming from irregular directions in a production unit) also fine?

What about the different sized fishes? What about the accuracy of models performed?

There are different varieties of fish species available. Some may have thick head while others may have thin head. Are these models providing the similar results in both the situations?

How this approach is useful for a fisherman, who is selling/cutting the fishes in a street market.

 Conclusions should be revised

References must be cross-checked to align with journal format

Reviewer 2 Report

Comments and Suggestions for Authors

Lines 31-32-37….: […] must be preceded by references of the form. All spelling written by word.

There is a problem with the files found in Figure 2.

Lines 235-236: Determined by these prices?

What is written in the graphs in Figure 8? But what inference do we draw from those graphs? These need to be explained.

Figure 9 should be explained in the same way.

Line 392: The results were too short. Just the results are given and it's done. But one more paragraph needs to be added. In this paragraph, it should be written about the passage limits, applicability to other fish, convertibility to mobile application, convertibility in other ways, and that success can increase if data is increased. Can it be used in industry and what is the benefit difference? The answers to these questions should be written down. Making the article better quality.

The entire article should be checked for spelling mistakes and language. There are also articles that are not suitable for the magazine format. For example, headings before and after headings.

These comments must be made. After it is made, the product will be of better quality.

Comments on the Quality of English Language

Minor corrections are required. Also correct spelling mistakes.
